The emerin-binding transcription factor Lmo7 is regulated by association with p130Cas at focal adhesions

Wozniak Michele A. 1 2
Baker Brendon M. 2
Chen Christopher S. 2
Wilson Katherine L. 1 klwilson@jhmi.edu
1 Department of Cell Biology, Johns Hopkins University School of Medicine , Baltimore, MD , USA
2 Department of Bioengineering, University of Pennsylvania , Philadelphia, PA , USA
Haraguchi Tokuko
Electronic publication date: 2013 Aug 20
Publication date: 2013
Volume: 1
Electronic Location ID: e134
Received 2013 Apr 14; Accepted 2013 Jul 29
Copyright: © 2013 Wozniak et al.
Copyright year: 2013
Copyright holder: Wozniak et al.
License: This is an open access article distributed under the terms of the Creative Commons Attribution License, which permits unrestricted use, distribution, and reproduction in any medium, provided the original author and source are credited.
License URL: https://creativecommons.org/licenses/by/3.0/

Keywords: Lmo7, p130Cas, Focal adhesions, Emery-Dreifuss muscular dystrophy, Laminopathy, LEM-domain, Nuclear envelope, Nucleoskeleton, Tendon, Emerin

Funding: National Institutes of Health grants postdoctoral fellowship NRSA F32 AR054219-01 postdoctoral fellowship NRSA F32 EB014691 RO1 GM74048 and RO1 EB00262 RO1 GM048646 This work was funded by the National Institutes of Health grants to MAW (postdoctoral fellowship NRSA F32 AR054219-01), BMB (postdoctoral fellowship NRSA F32 EB014691), CSC (RO1 GM74048 and RO1 EB00262) and KLW (RO1 GM048646). The funders had no role in study design, data collection and analysis, decision to publish, or preparation of the manuscript.

==============================
Loss of function mutations in the nuclear inner membrane protein, emerin, cause X-linked Emery-Dreifuss muscular dystrophy (X-EDMD). X-EDMD is characterized by contractures of major tendons, skeletal muscle weakening and wasting, and cardiac conduction system defects. The transcription factor Lmo7 regulates muscle- and heart-relevant genes and is inhibited by binding to emerin, suggesting Lmo7 misregulation contributes to EDMD disease. Lmo7 associates with cell adhesions and shuttles between the plasma membrane and nucleus, but the regulation and biological consequences of this dual localization were unknown. We report endogenous Lmo7 also associates with focal adhesions in cells, and both co-localizes and co-immunoprecipitates with p130Cas, a key signaling component of focal adhesions. Lmo7 nuclear localization and transcriptional activity increased significantly in p130Cas-null MEFs, suggesting Lmo7 is negatively regulated by p130Cas-dependent association with focal adhesions. These results support EDMD models in which Lmo7 is a downstream mediator of integrin-dependent signaling that allows tendon cells and muscles to adapt to and withstand mechanical stress.

Introduction

Lim domain only 7 (Lmo7) is a transcription factor with major roles in muscle, heart and other tissues (Ott et al., 2008) including lung epithelium, where Lmo7 is proposed to function as a tumor suppressor (Ott et al., 2008; Tanaka-Okamoto et al., 2009). Lmo7 regulates breast cancer cell migration by acting synergistically with the small GTPase RhoA to reduce G:F-actin ratios, leading to the activation of myocardin-related transcription factor (MRTF; also known as MAL or MKL1), a serum response factor (SRF) cofactor that activates cytoskeletal genes (Hu et al., 2011; Ho et al., 2013). Lmo7 shuttles into and out of the nucleus (Holaska, Rais-Bahrami & Wilson, 2006), and positively regulates many muscle- and heart-relevant genes (Holaska, Rais-Bahrami & Wilson, 2006; Ott et al., 2008). Among these genes is EMD which encodes a conserved nuclear membrane protein named emerin (Berk, Tifft & Wilson, 2013). Emerin, a LEM-domain protein, directly binds to membrane components of ‘LINC’ complexes (SUN-domain proteins, nesprins; Mislow et al., 2002; Zhang et al., 2005; Haque et al., 2010) and to the nucleoskeletal proteins lamin A and actin (Lee et al., 2001; Holaska et al., 2003; Holaska, Kowalski & Wilson, 2004). However emerin also directly binds transcription regulators including β-catenin (Markiewicz et al., 2006) and—notably—Lmo7 itself (Holaska, Rais-Bahrami & Wilson, 2006). In emerin-downregulated cells, Lmo7 nuclear localization is decreased or undetectable (Holaska, Rais-Bahrami & Wilson, 2006). In cells subjected to external mechanical force, emerin is required to activate specific ‘mechano-sensitive’ genes in response to force (Lammerding et al., 2005).

Loss of emerin, mutations in emerin-associated proteins (A-type lamins, nesprin-1, nesprin-2, LUMA) or mutations in transcription factor FHL1 are all genetically linked to Emery-Dreifuss muscular dystrophy (EDMD) (Meinke, Nguyen & Wehnert, 2011). These ‘EDMD genes’ suggest proper functioning of the affected tissues (heart, cardiac conduction system, specific skeletal muscles, major tendons) requires an emerin-containing multi-protein complex at the nuclear envelope (Simon & Wilson, 2011). Lmo7 is required for heart development in zebrafish, including development of the cardiac conduction system (Ott et al., 2008). The transcription-activator role of Lmo7 is inhibited by binding emerin, suggesting the emerin protein negatively feedback-regulates Lmo7 activity in the nucleus (Holaska, Rais-Bahrami & Wilson, 2006; Dedeic et al., 2011).

As a binding partner for emerin, Lmo7 was of particular interest because it localizes at cell–cell adhesions (Ooshio et al., 2004; Yamada et al., 2004), and might therefore transmit adhesion signals to the nucleus. Previous work suggested Lmo7 might also localize at focal adhesions, since a polypeptide comprising the C-terminal half of human Lmo7 (‘hLmo7C’; residues 888–1683) was detected both at the nuclear envelope and cell surface, where it co-localized with paxillin, when expressed transiently in HeLa cells (Holaska, Rais-Bahrami & Wilson, 2006). We report that endogenous Lmo7 associated with focal adhesions in both HeLa cells and mouse embryonic fibroblasts (MEFs), and co-immunoprecipitated with p130Cas, a major scaffolding and signaling component of focal adhesions (Defilippi, Di Stefano & Cabodi, 2006) that also influences myogenic differentiation (Kawauchi et al., 2012). The nucleocytoplasmic distribution of Lmo7, and the expression of six (of nine tested) Lmo7-regulated genes, were significantly altered in p130Cas-null MEFs. These results suggest Lmo7 activity is regulated by p130Cas-dependent association with focal adhesions. These findings are discussed in the light of a Drosophila study that showed A-type lamins exert their critical function in tendon cells (Uchino et al., 2013), which connect to muscle cells via the extracellular matrix, and evidence that integrin-dependent signaling is important for cells to respond to and withstand mechanical stress (Pines et al., 2012).

Materials and Methods

Cell culture and transfections

HeLa cells, wildtype MEFs and p130Cas-/- MEFs were maintained in 10% FBS in DMEM. HeLa cells were transiently transfected to express GFP (eGFP-C2; Clontech), GFP-rLmo7a, pcDNA3.1 myc, or myc-p130Cas using Lipofectamine PLUS (Invitrogen) per manufacturer instructions. The GFP-rLmo7a construct was a gift from Y. Takai (Osaka University). HeLa cells were obtained from C. Machamer (Johns Hopkins School of Medicine). Wildtype and p130Cas-/- MEFs and Myc constructs were gifts from P. Keely (University of Wisconsin-Madison). Mammalian cells were used under Johns Hopkins University Institutional Review Board approval (#B0807070104).

Indirect immunofluorescence and microscopy

To stain focal adhesions, cells were plated on fibronectin-coated coverslips for 30 min, two hours or six hours, then fixed (3.7% formaldehyde, 15 min), quenched (0.15 M glycine, 10 min), made permeable (0.2% Triton X-100; 10 min), blocked (1% bovine serum albumin in PBS; 1 h, 22–25°C) and incubated with primary antibodies (30 min, 22–25°C) specific for Lmo7 (H00004008-A01 from Novus, or HPA020923 from Sigma), vinculin (h-VIN-1; Sigma) or pFAK (4424G; Invitrogen), each at 1:100 dilution, or antibodies specific for p130Cas (06-500; Millipore; 1:20 dilution). Cells were rinsed with PBS and incubated 30 min (22–25°C) with DAPI (1:4000) plus either Alexa Fluor488- or Alexa Fluor555-conjugated anti-mouse or anti-rabbit secondary antibodies (1:200) from Invitrogen. Coverslips were rinsed in PBS and mounted on glass slides in PBS. For conventional epifluorescence imaging, coverslips were mounted in PBS with ProLong Gold Antifade reagent (Molecular Probes/Life Technologies, Grand Island, NY), and images were acquired using a Nikon Eclipse E600 microscope equipped with a Q-imaging Retiga EXi CCD camera and IPLab v3.9 software, or a Nikon TE200 microscope equipped with a Spot CCD camera and Spot software (Diagnostic Instruments). Total internal reflectance fluorescence (TIRF) microscopy was performed using a Nikon Eclipse Ti equipped with a CFI Apo TIRF 60x oil (N.A. 1.49) objective and Evolve EMCCD camera (Photometrics). To highlight colocalization, correlation images were created using a custom Matlab script: the intensities of two corresponding image channels at each pixel location were multiplied, and the resulting image was rescaled to its minimum and maximum values.

Generation of GST-fused domains of rLmo7a

The cDNA encoding each rLmo7a domain was amplified from a FLAG-rLmo7a construct (Ooshio et al., 2004; a gift from Y. Takai [Osaka University]) using the following primers, which included BamH1 and Xho1 restriction sites: CH domain primers were GGATCCGAGGCTCAGAGATGGGTGGAG and CTCGAGTTGTGCTTTTCTTCCCAGCCAGTA; F-Box primers were GGATCCCTACCTCCAGAAATCCAAGCGAAATTTCTC and CTCGAGAGTCAACATGTCGTCTTTCTTCAGTCG; PDZ domain primers were GGATCCCCCGGGACCAAACATGACTTTGG and CTCGAGTCCGTAGCGCCTGACATCC; and the LIM domain primers were GGATCCGTGTGCTCCTACTGTAATAGCATT and CTCGAGAGATTTGAATCGGAGATAGCAGTC. The resulting PCR products were ligated into the pGEM-T Easy vector (Promega) per manufacturer instructions. Ligation products were transformed into E. coli, and plasmids were purified. The pGEM-T-CH, -FBox, -PDZ or -LIM constructs were then excised by restriction with BamH1 and Xho1, ligated into the BamH1/Xho1-restricted pGEX 4T-3 vector for fusion to GST, and transformed into E. coli. Positive clones were identified by restriction analysis and verified by DNA sequencing.

Purification of GST-fused proteins

Transfected bacteria treated four hours with isopropyl β-D-1-thiogalactopyranoside (IPTG) were pelleted, resuspended in Tris-buffered saline (TBS) and sonicated. Triton X-100 was added to 1% (v/v) and lysates were rotated (15 min, 4°C). After centrifugation (12,000 rpm, 15 min, 4°C) we added 75 µl glutathione sepharose (Sigma) to the supernatant, then rotated (1 h, 4°C), briefly centrifuged to collect the beads, and washed three times in TBS.

Immunoprecipitations

Cells that transiently expressed GFP- or myc-tagged constructs for two days were lysed in Triple Detergent Lysis Buffer (50 mM Tris-Cl pH 8, 150 mM NaCl, 0.1% SDS, 1% NP40, 1% Triton X-100, Roche protease inhibitor complete mini tablet), and cleared by centrifugation. To immunoprecipitate GFP, lysates were incubated with 15 µl Protein A sepharose plus 2.5 µl GFP antibody (A6455, Molecular Probes) and rotated overnight (4°C). To immunoprecipitate Myc, cell lysates were incubated with 10 µl GammaBind G sepharose (Amersham Biosciences) plus 4 µg myc antibody (9E10, Santa Cruz) and rotated two hours (4°C). Bound proteins were collected by centrifugation, washed three times with Triple Detergent Lysis Buffer and eluted with SDS sample buffer. Proteins were resolved by SDS-PAGE, transferred to PVDF membrane and probed with antibodies against Lmo7 (CO5 or NO2, gifts from Y. Takai [Osaka University]; 1:5000), p130Cas (clone 21, BD Transduction Laboratories; 1:1000), talin (05-385, Millipore; 1:1000), emerin (serum 2999; Lee et al., 2001; 1:5000) or Myc (A-14, Santa Cruz; 1:1000). HRP-conjugated secondary antibodies (Jackson ImmunoResearch Laboratories; 1:5000) were detected by horseradish peroxidase chemiluminescence (Amersham).

GST pulldowns

HeLa cells were lysed in GST Lysis Buffer (100 mM HEPES pH 7.4, 150 mM NaCl, 2 mM EDTA, 0.1% SDS, 1% Triton X-100, 1 mM DTT, Roche protease inhibitor complete mini tablet), and cleared by centrifugation (12,000 g, 12 min, 4°C). The supernatant (lysate) was incubated (1.5 h, 4°C) with 15 µg recombinant GST, GST-CH, GST-F-Box, GST-PDZ or GST-LIM (see above). Beads were washed three times (GST Lysis Buffer); proteins were eluted with SDS sample buffer, resolved by SDS-PAGE, transferred to PVDF and probed with antibodies to GST (Santa Cruz Biotechnology, 1:1000), p130Cas (see above), or paxillin (BD Transduction Laboratories, 1:1000) as described above.

Cell fractionation

Cells were plated on fibronectin-coated petri dishes overnight, then fractionated using the NE-PER Kit (Thermo Scientific) per manufacturer instructions. Equal protein amounts (10 µg) of each fraction were resolved by SDS-PAGE and immunoblotted with antibodies to Lmo7 (Novus, 1:1000), β-tubulin (Sigma, 1:1000) or lamins A/C (Novacastra, 1:1000) as described above.

Quantitative real-time PCR

MEFs were rinsed with PBS and total RNA was extracted using the RNeasy mini kit (Qiagen). RNA (0.5 µg) was reverse transcribed using the high-capacity cDNA reverse transcription kit (Applied Biosystems) and the resulting cDNAs were amplified in an ABI 7300 system (Applied Biosystems). Results were analyzed using the ddCT method and normalized to 18S. Primers used were: 18s, GTAACCCGTTGAACCCCATT and CCATCCAATCGGTAGTAGCG; mef2D, CCTCAACAGTGCTAATGGAGCC and CCAAGTATCCAGCCGCATCC; Rbl1, GAATGCCTCTTGGATCTTTCC and GTGAACTTTCGAGGTGTTCCA; mef2C, ATGGGGAGAAAAAAGATTCAGATTACG and GCATGCGACTCTCTGAAGGATGGGC; Id2, ATGAAAGCCTTCAGTCCGGTGAGG and GCAAAGTACTCTGTGGCTAA; crebbp, TCCAGGGCGAGAATGTGACC and CCCTGTGCAGTCTCCACGGC; pcaf, AGCTGAACCCTCAGATCCCA and CACTTGTCAATCAACCCTGC; mbnl, ATGGCTGTTAGTGTCACACC and CTACATCTGGGTAACATACTTGTGGC; mef2B, ATGGGGAGGAAAAAAATCCAGATCTCAC and ACCGACATTGCGGGGGCCACGG; emerin, GTTTGCCTGCAATGGTACTGT and CAAGCACTTAAACCCATGAGC.

Results

To test potential focal adhesion localization of endogenous Lmo7, HeLa cells were cultured two hours on fibronectin-coated coverslips to allow focal adhesions to form, then double-stained by indirect immunofluorescence using antibodies specific for either Lmo7, vinculin or activated (Y397-phosphorylated) focal adhesion kinase (‘pFAK’). Because nuclear signals (typically ∼0.2 µm from the cell surface) are not reliably detected by TIRF microscopy, cells were imaged using either epifluorescence microscopy (Fig. 1A) or TIRF microscopy (Fig. 1B). Colocalization signals were further visualized using heat maps generated by multiplying intensities across fluorescent channels (Fig. 1, ‘cross-correlation’; see Methods). Lmo7 was detected in the nucleus (Fig. 1A) as expected. Lmo7 also localized at pFAK- and vinculin-positive focal adhesion sites (Figs. 1A and 1B), and at structures near the cell surface that resembled actin stress fibers (Fig. 1B). To determine if Lmo7 focal adhesion localization changed over time, we also used TIRF microscopy to image HeLa cells cultured on fibronectin-coated coverslips for thirty minutes or six hours (Fig. 1C). Lmo7 focal adhesion localization signals were highest at thirty minutes, when Lmo7 colocalized strongly with vinculin (MA Wozniak, unpublished data) and with pFAK at focal adhesion puncta located distal to the cell edge (Fig. 1C). Lmo7 did not co-localize perfectly with either pFAK or vinculin, consistent with many other focal adhesion proteins (Kanchanawong et al., 2010). These results supported the hypothesis that endogenous Lmo7 can localize at focal adhesions.

Figure 1 Endogenous Lmo7 co-localizes with vinculin and pFAK at focal adhesions in HeLa cells.

HeLa cells plated on fibronectin-coated coverslips for two hours (A, B) or other times (30 min or six hours; C) were fixed and stained by indirect immunofluorescence for endogenous Lmo7 (green) plus endogenous vinculin or pFAK (red). Cells were imaged by epifluorescence (A) or TIRF (B, C) microscopy, and co-localization was highlighted by cross-correlation analysis. Scale bars, 10 µm. Insets show each white-boxed region at higher magnification.

GFP-Lmo7a association with candidate focal adhesion proteins

The full Lmo7 polypeptide includes four homology domains (CH, F-box, PDZ and LIM domains; Fig. 2A) (Ooshio et al., 2004), some of which were candidate mediators of binding to focal adhesions. For example, LIM domains in other proteins can form homo- or hetero-dimers (Dawid, Breen & Toyama, 1998), and several resident focal adhesion proteins either have a LIM domain (e.g., paxillin (Turner & Miller, 1994) and zyxin (Sadler et al., 1992)), or bind to LIM-domain proteins (e.g., talin, which binds the LIM-domain of muscle protein NRAP (Luo, Herrera & Horowits, 1999), and p130Cas, which binds the LIM-domain of zyxin (Yi et al., 2002)). We tested potential Lmo7 association with three candidate endogenous focal adhesion proteins—talin, paxillin, and p130Cas—in HeLa cells that transiently expressed either GFP or GFP-fused full-length rat Lmo7 splicing variant a (GFP-rLmo7a; Fig. 2A), which is 71.8% identical to human Lmo7. Whole cell protein lysates were immunoprecipitated using GFP antibodies, resolved by SDS-PAGE and immunoblotted with antibodies specific for endogenous talin, paxillin or p130Cas. The endogenous proteins were each detected in input lysates (“I”; Fig. 2B; 5% loaded; talin not shown), and did not co-immunoprecipitate with GFP alone (Fig. 2B). GFP-rLmo7a showed no detectable association with talin (unpublished observations); however it co-immunoprecipitated weakly with paxillin and robustly with endogenous p130Cas (Fig. 2B, “P”; 80% loaded; n = 4), suggesting Lmo7 associates with p130Cas in vivo.

Figure 2 Lmo7 association with focal adhesion protein p130Cas.

(A) Schematic of the rat Lmo7a polypeptide and GFP- or GST-fused constructs used in this study. CH, predicted Calponin Homology domain; F-BOX, predicted F-box domain; PDZ, predicted PSD95/Dlg1/Zo-1 domain; NLS, predicted nuclear localization signal; LIM, LIM-domain. Boxes above rLmo7a indicate regions sufficient for direct binding to α-actinin (Ooshio et al., 2004), afadin (Ooshio et al., 2004) or emerin (Holaska, Rais-Bahrami & Wilson, 2006). (B) Whole cell protein lysates from HeLa cells that expressed GFP or GFP-rLmo7a for two days were immunoprecipitated with GFP antibodies, resolved by SDS-PAGE and immunoblotted with antibodies specific for paxillin or p130Cas. I, input (5% loaded); P, pellet (80% loaded). (C) Purified recombinant GST-fused Lmo7 polypeptides, resolved by SDS-PAGE and stained with Coomassie. (D–E) GST-pulldowns from whole HeLa cell lysates. Each GST-fused Lmo7 polypeptide (GST-CH, GST-F-box, GST-PDZ, GST-LIM), or GST alone, was incubated with HeLa cell lysates, then bound to glutathione, washed, eluted and resolved by SDS-PAGE. Bound proteins were detected in separate gels that were either stained with Coomassie (D), or immunoblotted for p130Cas (E) or paxillin (F). The black boxes in (D) indicate GST-fused proteins that migrated as SDS-resistant dimers.

Paxillin and p130Cas association with specific Lmo7 domains

To independently test candidate interactors, and potentially map binding region(s) within Lmo7, we fused GST to the N-terminus of the predicted CH, F-Box, PDZ or LIM domains of Lmo7 as depicted in Fig. 2A. Each purified recombinant polypeptide (Fig. 2C) was incubated with HeLa cell protein lysates, and glutathione-bound proteins were eluted with SDS-sample buffer and resolved in duplicate SDS-PAGE gels, which were either coomassie-stained (Fig. 2D) or immunoblotted for either endogenous p130Cas (130 kD; Fig. 2E) or paxillin (∼68 kDa; Fig. 2F). Qualitative inspection of coomassie-stained gels showed large amounts of each GST-fused ‘bait’, a low amount of each corresponding GST-dimer band (Fig. 2D, black squares), and additional unidentified bands; these included bands consistent with p130Cas (e.g., Fig. 2D, PDZ lane) and paxillin (e.g., Fig. 2D, F-box lane). Three different regions of Lmo7 (GST-CH, GST-PDZ, and GST-LIM) each consistently retained endogenous p130Cas (Fig. 2E, n = 3), but one—GST-PDZ—consistently retained the highest p130Cas signals. Paxillin was retained weakly by GST-LIM, and at high levels by GST-F-box (Fig. 2F; n = 3). These assays were qualitative, and did not distinguish between direct versus indirect binding to each Lmo7 fragment. Nevertheless, specific retention of paxillin by the F-box of Lmo7, and retention of p130Cas by three other domains (CH, PDZ, LIM), suggested that Lmo7 association with focal adhesions is mediated by association (direct or indirect) with paxillin and p130Cas. Further studies focused on p130Cas because it is a major focal adhesion scaffolding protein (Defilippi, Di Stefano & Cabodi, 2006) that binds zyxin, which (like Lmo7) shuttles to the nucleus (Nix et al., 2001).

Endogenous Lmo7 associates with p130Cas-myc and colocalizes with endogenous p130Cas in vivo

Lysates from HeLa cells that transiently expressed the empty Myc vector, or C-terminally Myc-tagged p130Cas (Cary et al., 1998), were precipitated using Myc antibodies and immunoblotted for endogenous Lmo7 (Fig. 3A). A large (∼200 kD) endogenous Lmo7 isoform co-immunoprecipitated consistently with Myc-p130Cas (Fig. 3A; n = 3). Furthermore, in both HeLa cells (n = 2) and MEFs (n = 3) plated on fibronectin-coated coverslips for two hours, indirect immunofluorescence double-staining showed a subset of endogenous Lmo7 co-localized with endogenous p130Cas in discrete puncta at focal adhesions, as visualized by epifluorescence microscopy (Fig. 3B) and TIRF imaging (Fig. 3C). We concluded Lmo7 associates with p130Cas at focal adhesions.

Figure 3 p130Cas associates with Lmo7 in HeLa cells and MEFs.

(A) HeLa cells that transiently expressed Myc-tagged p130Cas were immunoprecipitated with Myc antibodies and immunoblotted for endogenous Lmo7. (B, C) HeLa cells or MEFs were plated on fibronectin-coated coverslips two hours, fixed and double-stained by indirect immunofluorescence for endogenous Lmo7 (green) and endogenous p130Cas (red), then imaged by epifluorescence (B) or TIRF microscopy (C). Scale bars, 10 µm. Insets show each white-boxed region at higher magnification.

Lmo7 localization and gene regulation in p130Cas-/-cells

To determine if p130Cas influenced the nucleocytoplasmic distribution of Lmo7, we localized endogenous Lmo7 in wildtype versus p130Cas-/- MEFs (Honda et al., 1998). Indirect immunofluorescence staining and epifluorescence of p130Cas-/- MEFs revealed little or no detectable Lmo7 at the cell surface, and substantially higher nuclear signals, compared to wildtype MEF controls (Fig. 4A; n = 3). There were greatly reduced, but detectable, signals for endogenous activated (Y397-phosphorylated) FAK (‘pFAK’) at the cell surface and cytoplasm (Fig. 4A, α-pFAK). This suggested p130Cas was important, but not essential, for pFAK localization. The altered subcellular distribution of Lmo7 observed by epifluorescence was independently verified by cell fractionation and immunoblotting. Wildtype and p130Cas-/- MEFs were fractionated to separate nuclei from cytoplasm, and protein lysates were resolved by SDS-PAGE and probed with antibodies specific for Lmo7, A-type lamins (nuclear marker) or β-tubulin (cytoplasmic marker; Fig. 4B, n = 3). The nuclear and cytoplasmic markers were each enriched in the appropriate fraction (Fig. 4B). Densitometry and quantification of the nuclear-to-cytoplasmic signal ratio for Lmo7, relative to wildtype controls, confirmed the predominantly nuclear distribution of Lmo7 in p130Cas-/- cells (Fig. 4C). The difference in signal ratios was significant (p < 0.05 by the paired t-test; n = 3). We concluded that p130Cas regulates the subcellular distribution of Lmo7, and might normally retain Lmo7 outside the nucleus.

Figure 4 p130Cas regulates Lmo7 localization and Lmo7-dependent transcription.

(A) Indirect immunofluorescence images of wildtype (control) and p130Cas null (−/−) MEFs plated on fibronectin for two hours, then fixed and stained for endogenous Lmo7 (green) and pFAK (red). Scale bars, 10 µm (1 µm in insets; boxed). (B) Immunoblot of nuclear versus cytoplasmic fractions of wildtype (control) and p130Cas null (−/−) MEFs, resolved by SDS-PAGE and probed with antibodies specific for Lmo7, A-type lamins (nuclear marker), or β-tubulin (n = 3). (C) Quantification of the nuclear-to-cytoplasmic ratio of Lmo7 in wildtype (control) and p130Cas-null (−/−) MEFs (*p < 0.05, paired t-test, n = 3). (D) Quantitative real-time PCR analysis of mRNAs from genes known to be activated by Lmo7 (Mef2C, Id2, Crebbp, Pcaf, Mbnl, Mef2B, emerin) or repressed by Lmo7 (Mef2D, Rbl1), in control or p130Cas(-/-) MEFs. *p < 0.05 by the paired t-test; n = 4.

p130Cas influences Lmo7-dependent gene regulation

To determine if p130Cas-dependent Lmo7 localization was biologically relevant to genes regulated by Lmo7, we used quantitative real-time PCR to measure the mRNA levels of two genes (encoding Mef2D and Rbl1) that are negatively regulated by Lmo7, and seven genes (encoding Id2, Crebbp, PCAF, Mbnl, Mef2B, Mef2C, Emerin) positively regulated by Lmo7 (Holaska, Rais-Bahrami & Wilson, 2006). Relative to wildtype MEF controls, the mRNA levels of five Lmo7-activated genes (Id2, Crebbp, Pcaf, Mbnl and Mef2B) were significantly higher in p130Cas-/- MEFS (Fig. 4D; n = 4; p < 0.05 and Fig. S1). The magnitude of this increase ranged from 40% (Crebbp) to 70% (Pcaf, Mef2B) to 350%–400% (Id2, Mbnl). Of the Lmo7-inhibited genes, mRNA levels of one (Rbl1) were unaffected, whereas the other (Mef2D) decreased significantly (by 70%) relative to wildtype MEFs (Fig. 4D; n = 4; p < 0.05 by paired t-test and Fig. S1). Thus, six of nine tested genes responded in a manner consistent with higher Lmo7 activity in the nucleus of p130Cas-/- MEFs. These results demonstrated p130Cas is biologically relevant to the nuclear activity of Lmo7.

Discussion

Focal adhesion signaling regulates gene expression through mechanisms that remain unclear. Some focal adhesion proteins activate MAP kinase signaling and downstream gene expression (Howe, Aplin & Juliano, 2002). By contrast, other focal adhesion components transmit signals by directly translocating to the nucleus. These ‘direct translocators’ include zyxin, paxillin, Crp, FHL3 and Abl, all of which have a LIM domain(s) (Hervy, Hoffman & Beckerle, 2006), and most of which influence transcription (Krcmery et al., 2010). Some including nTrip6, paxillin and Hic5 are transcriptional co-activators. Nuclear Abl, a nonreceptor tyrosine kinase, has many roles in the nucleus (Hervy, Hoffman & Beckerle, 2006) and can also phosphorylate emerin directly in vitro (Tifft, Bradbury & Wilson, 2009). Our finding that endogenous Lmo7 localizes at focal adhesions, and is negatively regulated by association with p130Cas, coupled to evidence that Lmo7 is a ‘shuttling’ transcription factor (Holaska, Rais-Bahrami & Wilson, 2006), strongly supports the hypothesis that Lmo7 transmits signals from focal adhesions to the nucleus.

Lmo7 might resemble other transcription-regulating PDZ-LIM proteins, for which sequestration in the cytoplasm is important to ‘fine-tune’ cell- and tissue-specific activity in neurons (Kurooka & Yokota, 2005; Lasorella & Iavarone, 2006) and the heart (Camarata et al., 2010; Krcmery et al., 2010). Since most Lmo7 distributes throughout the cytoplasm and can also associate with cell adhesions, our findings suggest Lmo7 is dynamically regulated by a p130Cas-scaffolded focal adhesion kinase(s) or other signaling component(s) as it ‘cycles’ on and off focal adhesions.

Epifluorescence images (Fig. 3B) suggested p130Cas also localizes in the nucleus. However this localization is debated. For example GFP-p130Cas localizes at high levels in the nucleus (Kim et al., 2004), and polyclonal antibodies raised against GST-fused p130Cas residues 318–486 or 670–896 also stain the nucleus (Harte et al., 1996). Monoclonal antibody 4F4 detected nuclear signals in primary chicken embryonic cells; after Src-mediated transformation this signal localized at the cell surface (Kanner et al., 1991). Two studies suggest nuclear-localized p130Cas is less phosphorylated, hence potentially inactive (Kanner et al., 1991; Petch et al., 1995). However other antibodies do not detect p130Cas in the nucleus (e.g., Harte et al., 1996). Thus the nuclear localization and potential function of nuclear p130Cas are open questions.

How might p130Cas-dependent focal adhesion regulation of Lmo7 relate to Emery-Dreifuss muscular dystrophy (EDMD)? The major clinical aspect of EDMD is its affect on the heart (cardiomyopathy with potentially lethal cardiac conduction system defects), with major tendons and a subset of skeletal muscles also affected (Emery, 1987). Lmo7 is required for mouse C2C12 myoblasts to differentiate (Dedeic et al., 2011). Lmo7 activates muscle-specific genes such as myoD and pax3 early in differentiation; later (after myotube formation) Lmo7 localizes predominantly in the cytoplasm (Dedeic et al., 2011). Because sustained activation of pax3 inhibits myogenic differentiation (Boutet et al., 2007), we speculate that Lmo7 localization during muscle differentiation might be ‘fine-tuned’ by focal adhesion/p130Cas-dependent sequestration in the cytoplasm.

Our study did not reveal which domain(s) of p130Cas associate (directly or indirectly) with Lmo7, or, more importantly, how their association is regulated. However, our findings are consistent with a recent focal adhesion proteome study that reported myosin II-dependent recruitment of Lmo7 to focal adhesions in human foreskin fibroblasts (Kuo et al., 2011). Since focal adhesions grow and integrin-cytoskeleton connections are strengthened in response to myosin-mediated contractility and mechanical force (Chrzanowska-Wodnicka & Burridge, 1996; Choquet, Felsenfeld & Sheetz, 1997), we propose Lmo7 is involved in mechanical force-induced signaling. Supporting this idea, cell migration—which is strongly regulated by mechanically-induced signals (Lo et al., 2000)—is disrupted by loss of emerin (Emerson et al., 2009), p130Cas (Honda et al., 1998) or Lmo7 (Hu et al., 2011).

The proposed mechanotransduction function of Lmo7 might, we speculate, provide physiological feedback regulation of muscle contraction. Interestingly, p130Cas is one of several proteins that undergo physical extension in response to mechanical force (Sawada et al., 2006; Johnson et al., 2007; Grashoff et al., 2010). In fibroblasts, mechanical stretch is proposed to extend the substrate domain of p130Cas to expose fifteen YXXP motifs for phosphorylation by Src family kinases (Sawada et al., 2006). Tyrosine phosphorylation recruits signaling proteins such as Crk (Sakai et al., 1994), Nck (Schlaepfer, Broome & Hunter, 1997), and PTP-PEST (Garton et al., 1997), which then trigger downstream pathways involved in cell adhesion, migration and proliferation. Whether Lmo7 is influenced by stretch-induced phosphorylation of p130Cas is a new question raised by our findings.

p130Cas-dependent regulation of Lmo7 may be relevant to EDMD heart defects, since both proteins are important in the heart. In zebrafish, Lmo7 knockout leads to cardiac conduction system defects, including arrhythmia (Ott et al., 2008). In mice, p130Cas knockout is lethal during embryogenesis, with defects in cardiovascular development; cardiomyocytes have disorganized myofibrils and disorganized Z-disks (Honda et al., 1998), the cell surface structures that mechanically link the sarcomeres (contractile networks) of neighboring cardiomyocytes. Our discovery that p130Cas negatively regulates Lmo7 suggests the cardiovascular phenotypes of p130Cas-null mice arise at least in part from excess nuclear Lmo7 and consequent misregulation of Lmo7-dependent genes. In our study, loss of p130Cas affected six (of nine tested) Lmo7-regulated genes in a manner consistent with excess Lmo7 transcriptional activity in the nucleus. The three exceptions were mRNAs encoding Mef2C and emerin (which failed to increase in p130Cas-null MEFs) and Rbl1, which remained high in p130Cas-null MEFs. Further work is needed to understand why these genes ‘resisted’ transcriptional control by excess Lmo7 in MEFs. However we speculate that Lmo7 control over these ‘resistant’ genes might require a second p130Cas-dependent focal adhesion signaling event; possibilities include emerin phosphorylation by activated Abl or Src (Tifft, Bradbury & Wilson, 2009). Indeed Src phosphorylates a region of emerin required to bind Lmo7 and other transcription regulators (Tifft, Bradbury & Wilson, 2009).

The third EDMD-affected tissue, tendons, was recently studied in a Drosophila EDMD model with mutations in the A-type lamin (lamin ‘C’). Remarkably, the critical function of lamin C is exerted in tendon cells, not muscle, and involves the spectraplakin-dependent stabilization of the cytoskeleton (Uchino et al., 2013). Mammalian tendons are maintained by fibroblasts, which actively migrate and proliferate in response to injury (Arnesen & Lawson, 2006). Bone marrow mesenchymal stem cells can be triggered to differentiate into tendon cells (‘tenocytes’) by mechanical stretching, through a pathway that requires focal adhesion kinase and RhoA/ROCK (Xu et al., 2012). Stretch drives both focal adhesion growth (Shikata et al., 2005) and ROCK-mediated contractility (Xu et al., 2012), and stabilizes integrin-dependent force signaling at sites of muscle-tendon attachment (Pines et al., 2012). Hence our results identify Lmo7 as a downstream mediator of integrin-dependent signaling, and suggest that defects in focal adhesion signaling contribute to EDMD disease, particularly in tendons and muscle. Interestingly these results raise the possibility that a second p130Cas-dependent signal, speculated to involve Src or Abl, is required for Lmo7 to control a subset of genes, including those that encode emerin, Rbl1 and mef2C.

Supplemental Information

Figure S1 Raw data - quantitative real time PCR

Click here for additional data file.

The authors are grateful to Yoshimi Takai (Osaka University, Japan) for the rLmo7 constructs and Lmo7 antibodies, Patricia Keely (University of Wisconsin) for the wildtype and p130Cas(-/-) MEFs and myc constructs, and the Wilson Lab for helpful discussions.

Additional Information and Declarations

Competing Interests

Author Contributions

Ethics

Katherine L. Wilson is an Academic Editor for PeerJ.

Michele A. Wozniak conceived and designed the experiments, performed the experiments, analyzed the data, wrote the paper.

Brendon M. Baker conceived and designed the experiments, analyzed the data, contributed reagents/materials/analysis tools, wrote the paper.

Christopher S. Chen and Katherine L. Wilson conceived and designed the experiments, contributed reagents/materials/analysis tools, wrote the paper.

The following information was supplied relating to ethical approvals (i.e., approving body and any reference numbers):

The Institutional Review Board is Johns Hopkins University, and the IRB approval # (for HeLa, HEK293 and fibroblasts) is JHU IRB #B0807070104 (expires 6/30/2013).

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
