# Peer review of "The emerin-binding transcription factor Lmo7 is regulated by association with p130Cas at focal adhesions"

_PeerJ, doi:10.7717/peerj.134_

## Round 0.1 · original submission · Major Revisions

· Academic Editor

Major Revisions

This paper is very interesting and will provide important information to understand EDMD disease, thus it should be published soon. However, some of the image data are not clear as one of the reviewers suggested. For instance, Lmo7 and p130Cas in Figs. 3B and 3C are closely localized, but NOT co-localized. This paper will be greatly improved if you could provide better cell images as the reviewer suggest. I think this revision can be minor. But just in case you need time to get data enough to revise, I suggest "major" revision. I hope the reviewer's suggestions are helpful.

Reviewer 1 ·

Basic reporting

This is a well written and concise study that is both interesting and novel

Experimental design

1) Figure 1. The authors show focal adhesion staining of Lmo7 after 2 hours of cell spreading. Are the cells fully spread at this time point? It would be helpful to show an earlier and later time point (30mins and 6hrs) to confirm Lmo7 remains at focal adhesions throughout.

2) Figures 1, 3 and 4. In all these figures the authors show focal adhesion localisation. There is a lot of signal in the cell body. These images need to be more convincing. The quality of image could be improved greatly by using TIRF (Total Internal Reflection Fluorescence) microscopy. If not available, could confocal microscopy be used to image the bottom of the cells?

3) Figure 3C. The localisation of p130Cas looks nuclear as well as cytoplasmic. Is this true?

4) Figure 4. Does Lmo7 knockdown in the p130Cas null MEFs restore levels of Lmo7 targets to that of the control? This would confirm that Lmo7 is responsible for these changes.

5) Does Emerin over expression/knockdown impact on this pathway? Does emerin regulate the nuclear localisation of Lmo7? If true, increased emerin levels my prevent localisation of Lmo7 to focal adhesions.

6) Does Lmo7 knockdown, p130Cas disruption and emerin disruption interfere with cell adhesion and motility? This might be expected if these proteins are regulating cellular mechanics.

Validity of the findings

The findings are robust and conclusions are clearly stated.

Reviewer 2 ·

Basic reporting

No comments

Experimental design

No comments

Validity of the findings

No comments

Additional comments

Regulation of Lmo7, a transcriptional factor known to regulate muscle- and heart-relevant genes, has been considered to contribute in EDMD disease because emerin binding negatively regulate a transcriptional activity of Lmo7. Lmo7 is known to associate adhesions, and shuttle between the plasma membrane and nucleus, although physiological significance of such cellular localization of Lmo7 has not been shown. In order to dissect the possibility that Lmo7 is involved in transmitting adhesion signals to the nucleus, the authors examine subcellular localization of endogenous Lmo7 by IF, and show that it is associated with focal adhesions of cells. The authors further show that Lmo7 co-immunoprecipitates with p130Cas, a major scaffolding and signaling component of focal adhesions. Finally, the authors demonstrate by using p130Cas-null MEFs that nucleocytoplasmic distribution of Lmo7 shift towards cytoplasm, and expressions from genes (six out of nine genes examined) regulated by Lmo7 is altered (show higher Lmo7 activity) in the absence of p130Cas, suggesting that Lmo7 activity is negatively regulated through the association with p130Cas at focal adhesions. The results support the notion that the shuttling transcription factor Lmo7 transmit adhesion signals to the nucleus.
I think approaches used in the present manuscript are straightforward, experiments are performed carefully and presented clearly, and they are carefully discussed. I like an overall story.

---

## Round 0.2 · accepted · Accept

· Academic Editor

Accept

This is very nice paper important for understanding the molecular mechanisms of X-linked Emery-Dreifuss muscular dystrophy.